# Exploration of Resonant Modes for Circular and Polygonal Chladni Plates

**DOI:** 10.3390/e26030264

**Published:** 2024-03-15

**Authors:** Amira Val Baker, Mate Csanad, Nicolas Fellas, Nour Atassi, Ia Mgvdliashvili, Paul Oomen

**Affiliations:** 1The Works Research Institute, H-1044 Budapest, Hungary; 2Department of Atomic Physics, Eötvös Loránd University, H-1117 Budapest, Hungary

**Keywords:** Chladni plates, nodal lines, resonant modes, granular media, pattern generation, acoustic

## Abstract

In general, sound waves propagate radially outwards from a point source. These waves will continue in the same direction, decreasing in intensity, unless a boundary condition is met. To arrive at a universal understanding of the relation between frequency and wave propagation within spatial boundaries, we explore the maximum entropy states that are realized as resonant modes. For both circular and polygonal Chladni plates, a model is presented that successfully recreates the nodal line patterns to a first approximation. We discuss the benefits of such a model and the future work necessary to develop the model to its full predictive ability.

## 1. Introduction

Wave propagation in most dynamic systems is dependent on the energy of the system and the boundary conditions. Driven oscillatory wave systems provide an excellent environment for investigating wave propagation, resonant states, and the dependency on external factors, such as frequency and boundary conditions. Such investigations not only lead to greater insights into oscillatory systems in general but have also led to numerous advanced applications across a myriad of fields, e.g., seismology [1], quantum billiards [2], and nanomechanics [3,4].

One of the most extensively studied wave systems is that of nodal line patterns, formed by the excitation of a thin plate. In 1787, building on the observations of Galileo and Hooke, the musician and physicist Ernst Chladni demonstrated the various modes of vibration on a rigid surface [5,6,7]. These nodal patterns are now known as Chladni patterns or Chladni Figures, and the plates that are used as rigid surfaces are known as Chladni plates. Through the centuries, the type of surface and vibrating particle has taken many forms, such as from a brass plate and its shavings excited by a chisel [8]; a glass plate and flour excited by a violin bow string [9]; a metal plate and sand excited by a violin bow string [5]; a metal plate and sand excited by a block of high-density carbon dioxide [10,11,12,13]; to a drum membrane and quartz sand excited by singing through a pipe [14]. Now, the modern Chladni experiments are conducted using an electronically controlled mechanical oscillator that drives the plate at specific frequencies. The main difference is the accuracy of the driving frequency and the position of the driving source, which, for modern experiments, is typically the central position of the plate as opposed to the original experiments, which were generally driven by vibrating the edge of the plate. This difference in position alters the boundary conditions and, hence, the nodal line patterns.

To theoretically determine the nodal line patterns, several different forms and solutions of the wave equation have been used (e.g., Helmholtz, Bernoulli, etc.). For example, Kirchhoff [15,16,17] determined the vibration modes of a circular plate by considering the effects of deformation and stresses in a vibrating plate. This was then extended to the square plate for the case of free edges [16] and then for the case of clamped edges [16,18]. However, as noted by Wah [19], it is nearly impossible to simulate the case of ‘clamped edges’ in the laboratory and moreover plates generally behave as if they have boundary conditions between the theoretical ‘simple support’ and ‘clamped edge’ conditions. In any case, the response of the square plate was deemed of such a variable nature that determining an exact solution requires approximate or numerical procedures (e.g., Green’s function method, Ritz method, finite element analysis, etc.). In fact, exact solutions are only available for a few cases such as the circular plate which can be expressed in terms of the Bessel functions [20]. For more complex boundary conditions, the experimentally observed resonant modes do not correspond to theoretically determined eigenmodes; therefore, as noted by Tuan et al. [21], they should instead be solved from the inhomogeneous Helmholtz equation. Furthermore, as first noted by Waller [13,20,22,23], due to the effects of degeneracy and damping, the nodal line patterns at higher frequencies consist of two or more compounded nodes that need to be accounted for using different methods. This forming of the resonant mode by a superposition of numerous degenerate eigenmodes, or nearly degenerate eigenmodes, is also referred to as mode mixing (e.g., [24] and references therein).

To date, the most successful method in the analysis of nodal line patterns is that of the inhomogeneous Helmholtz equation. For example, utilising Green’s function, Tuan et al. [25] solved the inhomogeneous Helmholtz equation and found the response function for a vibrating wave on a thin plate as a function of the driving wave number, where the wave numbers are determined from the maximum entropy states. Thus, by substituting the theoretically determined wave numbers into the derived response function, the resonant modes can be calculated, and the experimental nodal line patterns can be successfully reconstructed for both the square and equilateral triangle plates [21,24,25,26,27,28]. Other approaches have also been successful; for example, instead of using the usual numerical approaches, Amore [29] found solutions to the Helmholtz equation (both homogeneous and inhomogeneous) by using the *little sinc* method [29,30,31]. Utilising this method, Amore [29] obtained a discretization of a finite region of space in the 2D plane and successfully calculated the modes of vibration for membranes of arbitrary shapes. It should be noted that this approach has not yet been applied to nodal line patterns on Chladni plates. However, although predictions of the resonant modes can be made utilising these approximation methods, they do not yield exact solutions that can deepen our understanding of the system dynamics and require techniques that do not necessarily translate across more complex boundary conditions.

In an effort to deepen our understanding and find meaningful solutions, we explore the resonant modes for circular and polygonal Chladni plates. A simple model is presented where the resonant modes (m = 1–5) are determined for the low-frequency range (<2 kHz) and for nodal lines n = 1–3. Using classical mechanics, and by considering a standing wave emanating from the centre of the plate, the resonant modes are defined as equilibrium points or maximal entropy states, which are given in terms of the wave number and the response function of the plate. The method presented considers the response function in terms of the geometry of the spatial boundaries and is thus potentially applicable to all boundary conditions.

## 2. Resonant Mode Chladni Patterns

Each Chladni plate, depending on the material it is made of, as well as its size and shape, will have resonant modes, i.e., frequencies at which standing waves are formed. Before we can investigate the behaviour of these modes, we need to determine the resonant frequencies. This can be achieved by investigating the impedance of the mechanical oscillator as a function of frequency.

### 2.1. Impedance Experiment to Determine Resonant Modes

Following the work of Tuan et al. [24], we determine the resonant mode frequencies by measuring the impedance of the mechanical wave driver, with and without a plate attached.

The setup for the experiment (shown in Figure 1) consists of a 20 W Thomann TA50 Amplifier (Thomann, Treppendorf, Germany) connected in series with a PASCO Mechanical Wave Driver (Arbor Scientific, Saline, MI, USA) and a GDM 8342 Multimeter (GW Instek, Conrad, Budapest, Hungary). The mechanical wave driver was driven by the amplifier at ~30% via a Max/MSP patch that generated a sinusoidal wave over the frequency range of 20–2000 Hz. The multimeter took measurements of the AC voltage every second over a period of ~20 min, yielding a resolution of ~1.6 Hz. These measurements were initially carried out with no plate attached to the mechanical wave driver. Subsequent measurements were then taken for a total of ten plates of the following shape and dimensions: a circle 18 cm and 24 cm in diameter; a triangle with sides measuring 18 cm and 24 cm; a square with sides of 18 cm and 24 cm; a pentagon with sides of 9.5 cm and 14.5 cm; a hexagon with sides of 9 cm and 12 cm. The plates were made of acrylic with a thickness of 2 mm, resulting in an aspect ratio ≪0.1 for all plates.

In each case, the impedance, *Z_f_,* was calculated as a function of frequency from the variable voltage output, *V_f_*, and the base current, *I*, via the formula *Z_f_* = *V_f_*/*I*. The effective impedance was then determined by subtracting the impedance, found in the case where no plate was attached, from the impedance found when the plate was attached. Figure 2a–e shows the results for each plate, where the impedance is shown as a function of frequency and the peaks indicate the resonant frequencies in each case. The resonant frequencies found from the impedance peaks are shown in Table 1.

### 2.2. Chladni Plate Vibration at Resonant Mode Frequencies

Following the setup shown in Figure 1, we investigated the patterns formed at the resonant frequencies determined from the impedance analysis (see Section 2.1). For each of the 10 plates, a sand sprinkler was used to completely cover the plate with sand. Once the desired frequency was set, more sand was added as necessary, ensuring optimum pattern formation. To record pattern formation, a Canon EOS 5D Mark IV Camera (220volt.hu, Budapest, Hungary), with a Canon 100 mm macro lens, was mounted parallel to the Chladni plates. The recordings were taken in video mode, with the aperture set at F2.8, the shutter speed set to 1/50, and the ISO set at 25600. Figure 3, Figure 4, Figure 5, Figure 6, Figure 7, Figure 8, Figure 9, Figure 10, Figure 11 and Figure 12 show the resulting resonant frequency pattern formations, also referred to as nodal line patterns, for each of the 10 plates. As expected, the number of nodal lines exhibited increases with increasing frequency, and, for the polygonal plates, the degree of complexity also increases with frequency. The same pattern morphology is exhibited in both the larger-sized plates and smaller-sized plates but is generally exhibited at a higher frequency in the smaller-sized plates. This observed difference in frequency between the larger and the smaller plates increases with increasing modes.

## 3. Theoretical Determination of the Nodal Line Patterns

To theoretically recreate the nodal line patterns for the resonant modes, we consider a simple wave equation for a standing wave. Let *z* represent the variable amplitude of the wave as it propagates up and down,
(1)$$z_{1}=Asin\;\left(\omega t\right)$$
(2)$$z_{2}=Asin\;\left(\omega t+\phi \right)\;$$
(3)$$z_{r}=z_{1}+z_{2}=2Acos\;\left(\frac{\phi }{2}\right)\;sin\;\left(\omega t+\frac{\phi }{2}\right)$$
here, *z*_1_ is the outward wave; *z*_2_ is the reflected wave; *z*_r_ is the resultant wave; *A* is the amplitude; *ω* is the angular frequency; and *ϕ* is the phase difference between the outward and reflected wave.

The nodal points, where the sand collects, correspond to the points of zero displacement, also described as equilibrium points or maximum entropy states, e.g., see ref. [28]. For each mode, these points can be found by setting *z*_r_ in Equation (3) to equal 0, i.e.,
$$z_{r}=2Acos\;\left(\frac{\phi }{2}\right)\;sin\;\left(\omega t+\frac{\phi }{2}\right)=0\;$$
we can then assume that,
$$sin\;\left(\omega t+\frac{\phi }{2}\right)=0\;$$
giving,
$$\omega t+\frac{\phi }{2}=n\pi$$
and, thus, the time it takes for the wave to reach a given point is given as,
(4)$$t=\frac{\left(\;n\pi \;\;\frac{\phi }{2}\;\right)}{\omega }$$
If the wave is reflected and oscillates at the resonant frequency of the system, then there is no phase difference between the waves; therefore, *ϕ* = 0, and a standing wave is formed. At specific frequencies, Chladni plates are known to exhibit coherent patterns referred to as Chladni Figures, or nodal line patterns. These nodal line patterns consist of nodal points which, like in Equation (4), can be expressed in terms of time,
(5)$$t=\frac{n\pi }{\omega }$$
In polar coordinates (*r*, *θ*, *φ*) at *z*_r_ = 0, *φ* = *π*/2 (note the polar coordinate *φ* is not the same as the phase difference *ϕ* defined earlier). We can, therefore, assume that *r* only exists in the *x*, *y* plane, which physically represents the spatial dimension of the plate. If we then consider the spatial dimension of the plate in terms of time, we can define *r* as,
$$r=t=\frac{n\pi }{\omega }$$
and we can then also set the boundary condition in terms of time as,
(6)$$t=L_{\theta }/v$$
where *L*_*θ*_ is the spatial dimension given as a function of *θ*; *t* is the time it takes for the wave to traverse the distance *L*_*θ*_; and *v* is the velocity of the wave. Note, the velocity of a sound wave is ~340 ms^−1^ in air (at average room temperature); ~1500 ms^−1^ in water; and, for solids, it is given as $v=\sqrt{\frac{Y}{\rho }}$, where $Y=\frac{\sigma \left(\varepsilon \right)}{\varepsilon }=\frac{F/A}{\Delta L/L_{0}}$ is the Young’s Modulus given in pascals (Pa); *σ*(*ε*) is a measure of stress; *ε* is a measure of strain; and *ρ* is the density. However, in the case of the Chladni plate, we are not interested in the velocity of the sound wave. Instead, we are interested in the velocity of the mechanical wave which lifts the sand grains up and down, which can be given as *v* = *λ**f*. For a circular plate of known diameter, the wavelength, and hence velocity, can be determined from the observed nodal line patterns. For the circular plate, the resulting velocity as a function of frequency was thus found to be,
(7)$$v=2\sqrt{2}f^{\frac{1}{2}}$$
This is in good agreement with the observations made by Tuan et al. [24] and was subsequently used to determine the velocity for each of the plates. 

Therefore, Equation (5) gives us a solution for the points of equilibrium, the nodal points that form the nodal patterns for each resonant mode. However, the plate is also subject to forces that will influence wave propagation and the motion of the sand particles. We know that the aspect ratio of the plate is ≪0.1; therefore, we can assume that the response function is variable in the *x*, *y* plane (*r*, *θ* plane) and negligible in the *z* plane (*φ* plane). To account for this, we can include a response function, *F*_*R*_, such that Equation (5) becomes,
(8)$$r=\frac{n\pi -F_{R}}{\omega }=kv\left(n\pi -F_{R}\right)$$
where k is the wave number defined in terms of the angular frequency, *ω*, and velocity, v. 

### 3.1. Circular Chladni Plate

To begin, we take the example of a circular Chladni plate, which has one boundary condition that is constant with respect to *θ*. In this example, we can define the response function, *F*_*R*_, as a simple coefficient, which we call ***C*** (see Table 2 for the list of coefficients). The nodal points making up the nodal line patterns are then determined from Equation (8) and by setting the velocity *v*, as given by Equation (7) and the boundary condition in terms of *L* = *a*/2 (where *a* is the diameter of the circular plate). In each case, the coefficient, ***C***, was first determined by manually adjusting the value until the theoretical nodal line patterns match those of the experiment. This was performed both visually and by using a fitting function that adjusts the coefficients according to the fit. Note that all equations and boundary conditions were implemented using a custom software application created in Python 3.10.

We find the resulting nodal line patterns (see Figure 13 and Figure 14) to be in good agreement with the experimental results. As well, it is interesting to note that, depending on the position of the nodal line relative to the plate centre, we see lesser or greater effects of the response function. For example, at a critical closeness to the centre, we see a greater response function. Then, as we move further away from the centre of the plate, we see a smaller response function, until reaching a critical distance away from the centre, where we, again, see almost zero response function. For a specific plate, this is clearly expressed in the value of the ***C*** coefficient, which increases with increasing frequency and decreases with an increasing number of nodal lines (see Table 2 for the full list of coefficients).

### 3.2. Square Chladni Plate

As with the circular plate, we expect the nodal points to be shifted due to the response function of the plate. However, as we are dealing with a polygon, we must also take into consideration the variability of *L*, which, for any n-sided polygon, varies with respect to the angle with the primary axis, *θ*. Thus, for a square plate, the minimum length from the centre is given as,
(9)$$L_{min}=a/2$$
where *a* is the length of one side and, thus, *L*_*θ*_ varies as,
(10)$$L_{\theta }=\left(\begin{array}{c} \frac{L_{min}}{\cos\theta },\theta \;\in \frac{-\pi }{4},\frac{\pi }{4}\\ \frac{L_{min}}{\sin\theta },\theta \;\in \frac{\pi }{4},\frac{3\pi }{4}\\ \frac{-L_{min}}{\cos\theta },\theta \;\in \frac{3\pi }{4},\frac{5\pi }{4}\\ \frac{-L_{min}}{\sin\theta },\theta \;\in \frac{5\pi }{4},\frac{7\pi }{4} \end{array}\right)$$
The response function, *F*_*R*_, for the square plate, or any other polygon, will thus depend on the variability in the length of the plate, with respect to the centre, *θ*, as well as the distance from the centre, *r*. To investigate this further, and as a first approximation, we set the response function in the form of a trigonometric function, e.g., a rose function,
(11)$$F_{R}(\theta ,r)=A\;cos\;\left(N\theta \right)+C$$
where N is the number of sides of the polygon; therefore, for the square plate, N = 4. Note, to distinguish between the nodal line number, *n*, we refer to the sides of the polygon as N.

From Equation (8), and by setting *F*_*R*_ as given by Equation (11), the boundary condition in terms of *L*_*θ*_ as given by Equation (10), and the velocity *v*, as given by Equation (7), we recreated the nodal line patterns in a good approximation with the experimental results (see Figure 15 and Figure 16). It is interesting to note that, depending on the position of the nodal line relative to the plate centre, we see lesser or greater effects of the response function variability. For example, at a critical closeness to the centre, we see no variability in the response function, and it acts in the same way as for a circular plate. Then, as we move further away, it exhibits more variability, until reaching a critical distance away from the centre, where again we again see no variability in the response function. For a specific plate, this is clearly expressed in the value of the ***A*** coefficient, which appears to oscillate between positive and negative values with the increasing number of nodal lines. As well as the circular plate, the value of the ***C*** coefficient increases with increasing frequency for a specific nodal line (defined as ***C***_r_) and decreases with an increasing number of nodal lines for a specific frequency (see Table 2 for the full list of coefficients). For the higher frequencies, we can see the beginnings of mode mixing, not yet fully effecting. As we move into increasingly higher frequencies, this will be more evident.

### 3.3. Triangular Chladni Plate

As with the square plate, we must also take into consideration the variability of *L* with respect to *θ*. Thus, for a triangular plate, the minimum length from the centre is given as,
(12)$$L_{min}=\frac{\frac{a}{2}}{tan\;\theta \;}$$
where *a* is the length of one side and, thus, *L*_*θ*_ varies as,
(13)$$L_{\theta }=\left(\begin{array}{c} \frac{L_{min}}{\cos\left(\theta -0\right)},\theta \;\in \frac{-\pi }{3},\frac{\pi }{3}\\ \frac{L_{min}}{\cos\left(\theta -\frac{2\pi }{3}\right)},\theta \;\in \frac{\pi }{3},\pi \\ \frac{L_{min}}{\cos\left(\theta -\frac{4\pi }{3}\right)},\theta \;\in \pi ,\frac{5\pi }{3} \end{array}\right)$$

From Equation (8), by setting *F*_*R*_ as given by Equation (11) with N = 3, the boundary condition in terms of *L*_*θ*_ as given by Equation (13), and the velocity *v*, as determined from the experimental analysis, as given by Equation (7), we recreate the nodal line patterns in a good approximation with the experimental results (see Figure 17 and Figure 18). It is interesting to note that, as with the square plate, we see the same progression of pattern complexity, with the value of the coefficients following a similar relationship to that of both the circle and the square plate.

### 3.4. Pentagon Chladni Plate

As with the other polygon plates, we must also take into consideration the variability of *L* with respect to *θ*. Thus, for a pentagon plate, the minimum length from the centre is given as,
(14)$$L_{min}=\frac{\frac{a}{2}}{tan\;\left(\frac{\pi }{5}\right)\;}$$
where *a* is the length of one side and, thus, *L*_*θ*_ varies as,
(15)$$L_{\theta }=\left(\begin{array}{c} \frac{L_{min}}{\cos\theta },\theta \;\in \frac{-\pi }{5},\frac{\pi }{5}\\ \frac{L_{min}}{\cos\left(\theta -\frac{2\pi }{5}\right)},\theta \;\in \frac{\pi }{5},\frac{3\pi }{5}\\ \frac{L_{min}}{\cos\left(\theta -\frac{4\pi }{5}\right)},\theta \;\in \frac{3\pi }{5},\pi \\ \frac{L_{min}}{\cos\left(\theta -\frac{6\pi }{5}\right)},\theta \;\in \pi ,\frac{7\pi }{5}\\ \frac{L_{min}}{\cos\left(\theta -\frac{8\pi }{5}\right)},\theta \;\in \frac{7\pi }{5},\frac{9\pi }{5} \end{array}\right)$$

From Equation (8), by setting *F*_*R*_ as given by Equation (11) with N = 5, the boundary condition in terms of *L*_*θ*_ as given by Equation (15), and the velocity *v*, as given by Equation (7), we recreate the nodal line patterns in a good approximation with the experimental results (see Figure 19 and Figure 20). It is interesting to note that, again, like the other polygon plates, we see the same progression of pattern complexity, with the value of the coefficients following a similar relationship to that of both the circle and the other polygon plates.

### 3.5. Hexagon Chladni Plate

Again, as with the other polygon plates, we must also take into consideration the variability of *L* with respect to *θ*. Thus, for a hexagon plate, the minimum length from the centre is given as,
(16)$$L_{min}=\frac{\frac{a}{2}}{tan\;\left(\frac{\pi }{6}\right)\;}$$
where *a* is the length of one side and, thus, *L*_*θ*_ varies as,
(17)$$L_{\theta }=\left(\begin{array}{c} \begin{array}{c} \frac{L_{min}}{\cos\theta },\theta \;\in \frac{-\pi }{6},\frac{\pi }{6}\\ \frac{L_{min}}{\cos\left(\theta -\frac{\pi }{3}\right)},\theta \;\in \frac{\pi }{6},\frac{3\pi }{6}\\ \frac{L_{min}}{\cos\left(\theta -\frac{2\pi }{3}\right)},\theta \;\in \frac{3\pi }{6},\frac{5\pi }{6}\\ \frac{L_{min}}{\cos\left(\theta -\pi \right)},\theta \;\in \frac{5\pi }{6},\frac{7\pi }{6}\\ \frac{L_{min}}{\cos\left(\theta -\frac{4\pi }{3}\right)},\theta \;\in \frac{7\pi }{6},\frac{9\pi }{6}\\ \frac{L_{min}}{\cos\left(\theta -\frac{5\pi }{3}\right)},\theta \;\in \frac{9\pi }{6},\frac{11\pi }{6} \end{array} \end{array}\right)$$

From Equation (8), by setting *F*_*R*_ as given by Equation (11) with N = 6, the boundary condition in terms of *L*_*θ*_ as given by Equation (17), and the velocity *v*, as given by Equation (7), we recreate the nodal line patterns in a good approximation with the experimental results (see Figure 21 and Figure 22). It is interesting to note that, again, like the other polygon plates, we see the same progression of pattern complexity, with the value of the coefficients following a similar relationship to that of both the circle and the other polygon plates.

## 4. Summary

In this paper, we analyse the experimentally measured resonant states of 10 different shaped and sized circular and polygonal Chladni plates and present a simple model that considers a response function in terms of the geometry of the spatial boundaries.

The results show that, depending on the position of the nodal line relative to the plate centre, we see lesser or greater effects on the response function. For example, at a critical closeness to the centre, we see a greater response function. Then, as we move further away from the centre of the plate, we see a smaller response function, until reaching a critical distance away from the centre, where we again see almost zero response function. For the polygon plates, these effects extend to the variability within nodal lines, where, at a critical closeness to the centre, we see no variability in the response function, and it acts in the same way as one would see for a circular plate. Then, as we move further away, it exhibits more variability, until reaching a critical distance away from the centre, where, again, we see no variability in the response function. This behaviour is expected, as, at a critical point, the boundary effects become either negligible or significant. For all plates, circular and polygonal, the value of the ***C*** coefficients, for a specific nodal line (defined as ***C***_r_), increases with increasing frequency and, for a specific frequency, decreases with an increasing number of nodal lines. For the polygonal plates, the value of the ***A*** coefficient, for a specific nodal line (defined as ***A***_r_), appears to oscillate between positive and negative values (or between higher and lower values), tending to zero as the frequency increases. Moreover, for a specific frequency, the ***A*** coefficient decreases with an increasing number of nodal lines. However, at the higher frequencies, there are a couple of exceptions to this relationship (i.e., the hexagon *a* = 12 cm, *f* = 1635 Hz; and the pentagon *a* = 14.5 cm, *f* = 1071 Hz). These exceptions are most likely due to the effects of mode mixing. Note, with the range of frequencies investigated in this study, we can only see the beginnings of mode mixing, which will become more evident as we investigate increasingly higher frequencies. For these effects to be accounted for and a quantitative relationship to be made, more investigations yielding a larger dataset are required and will be carried out in a future study. 

It is obvious that, for a non-idealized system, the resonance modes will not equate with the eigenmodes of the wave equation. Instead, the maximum entropy states, defined as the points of equilibrium, will be affected by the response function, which causes a shift and, hence, a discrepancy between the eigenmodes and the resonant modes. In addition, the boundary conditions and restrictions will result in mode mixing, resulting in further deviations from the eigenmodes. The resonant modes have successfully been determined by solving the inhomogeneous Helmholtz equation to find the response function in terms of the driving wave number, where the wave number is theoretically determined from the maximal entropy states, as determined from the standard Shannon equation for entropy [21,24,25,26,27,28]. However, in the study presented here, we determine the maximal entropy states as the points of equilibrium, defined in terms of the wave number and the response function, where the response function is defined in terms of the geometry of the spatial boundaries. The results presented here thus show that the deviations from the eigenmodes are not the result of inhomogeneity and are rather a coherent geometrical effect that determines the response function. This is not only evident from the symmetry of the nodal line patterns but also from the simplistic method presented to reconstruct the nodal patterns.

This study considers a finite range of frequencies and plate geometries and presents a simple model that has been shown to successfully recreate the nodal line patterns to a first approximation. However, more data are needed so that quantitative analysis can define the coefficients as a function of the driving frequency and boundary conditions and as well include the effects of mode mixing at higher frequencies. We envision that the development of this model to its full predictive ability is achievable and has huge potential for a wide range of applications. For example, with a model that defines the coefficients, the resonant modes could be determined for more complex and asymmetric boundary conditions. This could then be applied to different types of mediums as well as different points of excitation and multiple points of excitation and be extended to higher dimensions. Compared with more approximate methods, this simple model has the potential to improve both accuracy and efficiency, as well as deepen our understanding of system dynamics. Further investigations are, therefore, needed to extend this model to a broader range of frequencies, mediums, and more complex plate geometries.

## Figures and Tables

**Figure 1 entropy-26-00264-f001:**
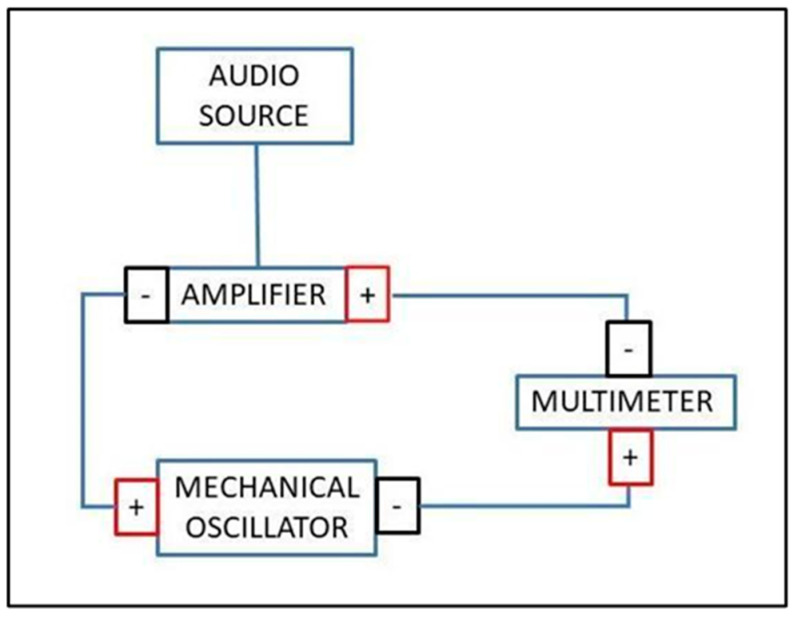
Schematic of experimental setup.

**Figure 2 entropy-26-00264-f002:**
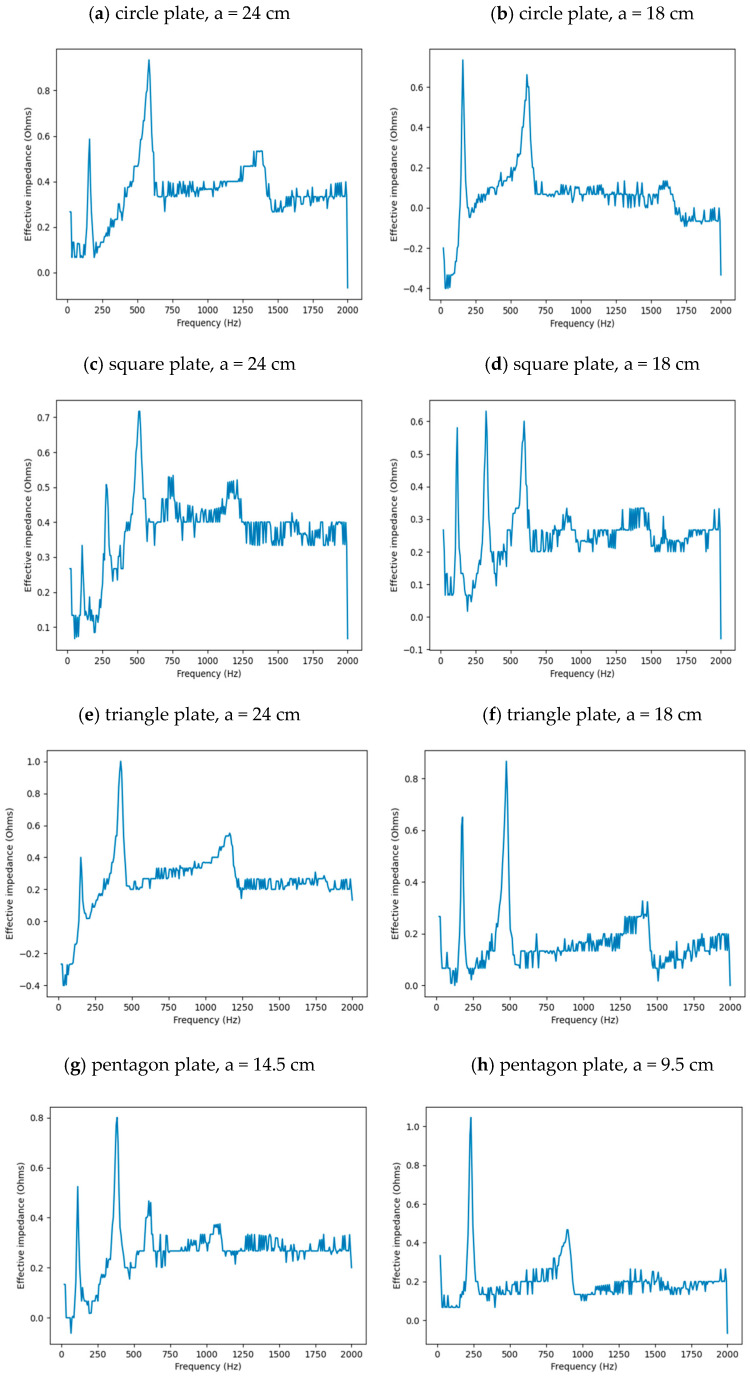
Plots of the effective impedance of a mechanical oscillator with a plate attached shown as a function of frequency for the following plates: (**a**) large circle; (**b**) small circle; (**c**) large square; (**d**) small square; (**e**) large triangle; (**f**) small triangle; (**g**) large pentagon; (**h**) small pentagon; (**i**) large hexagon; and (**j**) small hexagon. The peaks indicate the resonance frequencies which are listed in Table 1 below.

**Figure 3 entropy-26-00264-f003:**
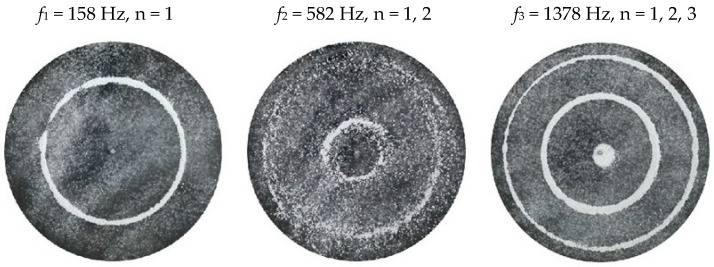
Nodal line patterns observed in the larger circular plate, a = 24 cm.

**Figure 4 entropy-26-00264-f004:**
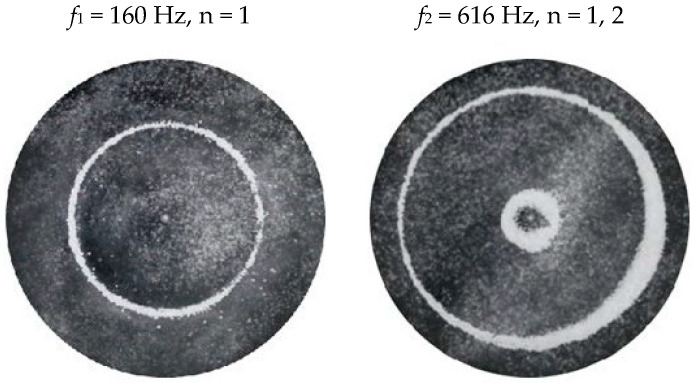
Nodal line patterns observed in the smaller circular plate, a = 18 cm.

**Figure 5 entropy-26-00264-f005:**
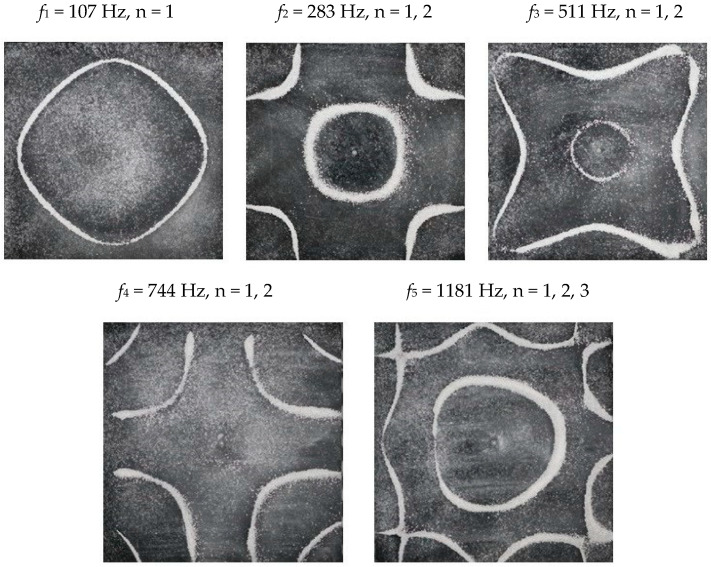
Nodal line patterns observed in the larger square plate, a = 24 cm.

**Figure 6 entropy-26-00264-f006:**
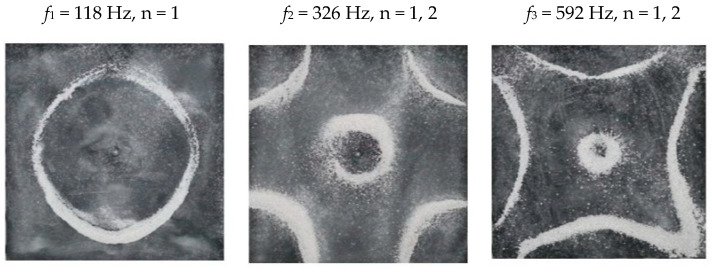
Nodal line patterns observed in the smaller square plate, a = 18 cm.

**Figure 7 entropy-26-00264-f007:**
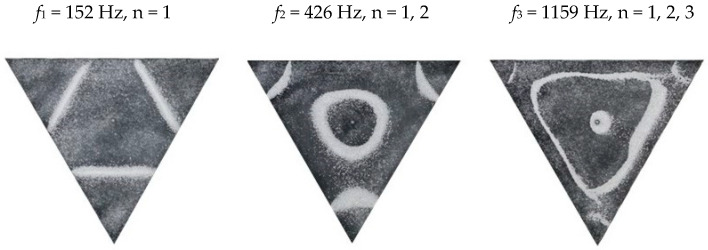
Nodal line patterns observed in the larger triangle plate, a = 24 cm.

**Figure 8 entropy-26-00264-f008:**
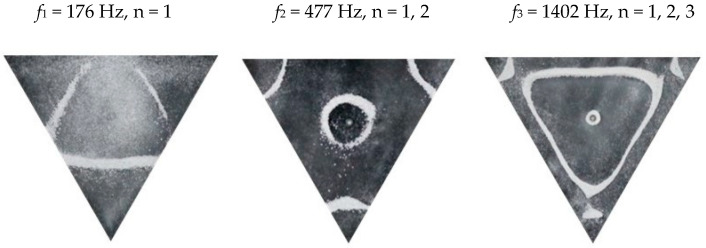
Nodal line patterns observed in the smaller triangle plate, a = 18 cm.

**Figure 9 entropy-26-00264-f009:**
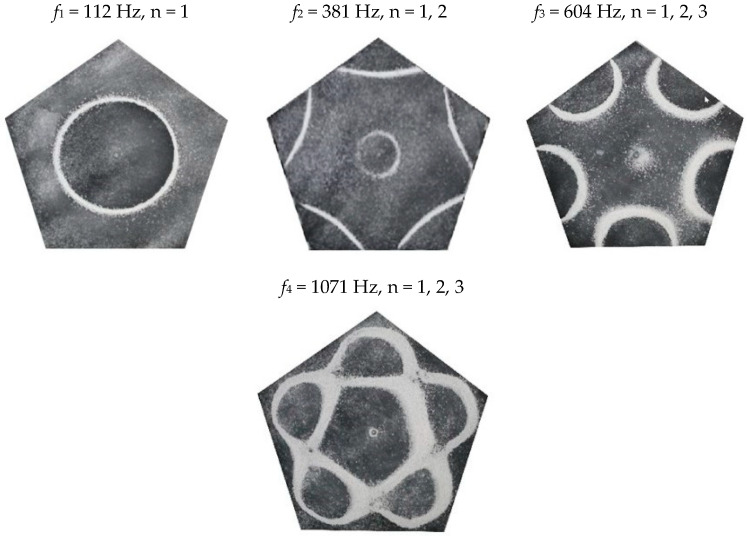
Nodal line patterns observed in the larger pentagon plate, a = 14.5 cm.

**Figure 10 entropy-26-00264-f010:**
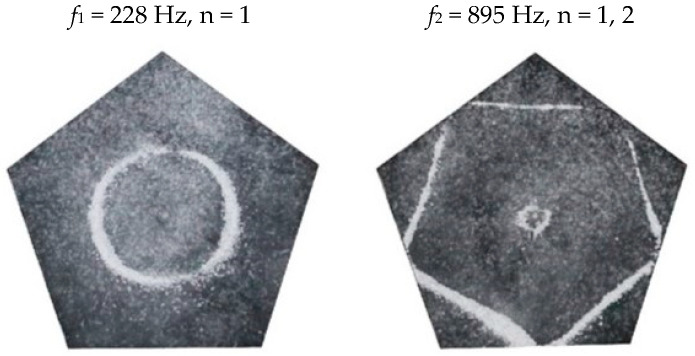
Nodal line patterns observed in the smaller pentagon plate, a = 9.5 cm.

**Figure 11 entropy-26-00264-f011:**
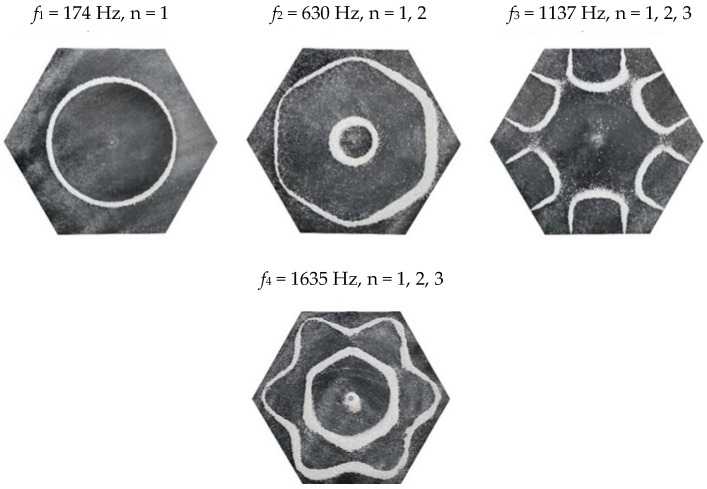
Nodal line patterns observed in the larger hexagon plate, a = 14.5 cm.

**Figure 12 entropy-26-00264-f012:**
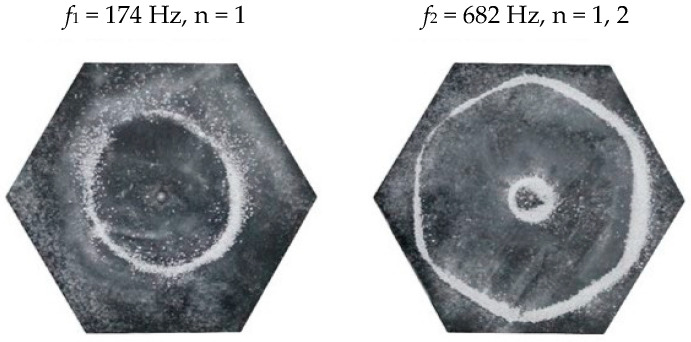
Nodal line patterns observed in the smaller hexagon plate, a = 9 cm.

**Figure 13 entropy-26-00264-f013:**
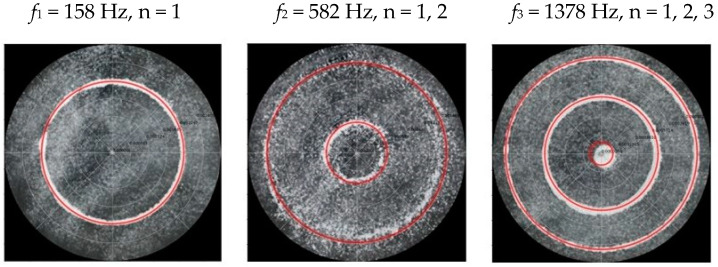
Simulation of nodal line patterns for the larger circular plate, a = 24 cm.

**Figure 14 entropy-26-00264-f014:**
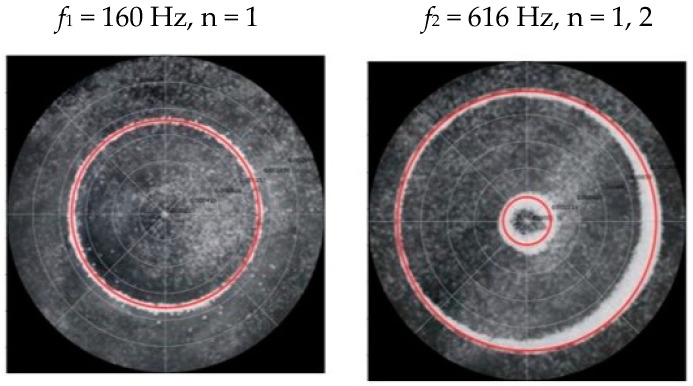
Simulation of nodal line patterns for the smaller circular plate, a = 18 cm.

**Figure 15 entropy-26-00264-f015:**
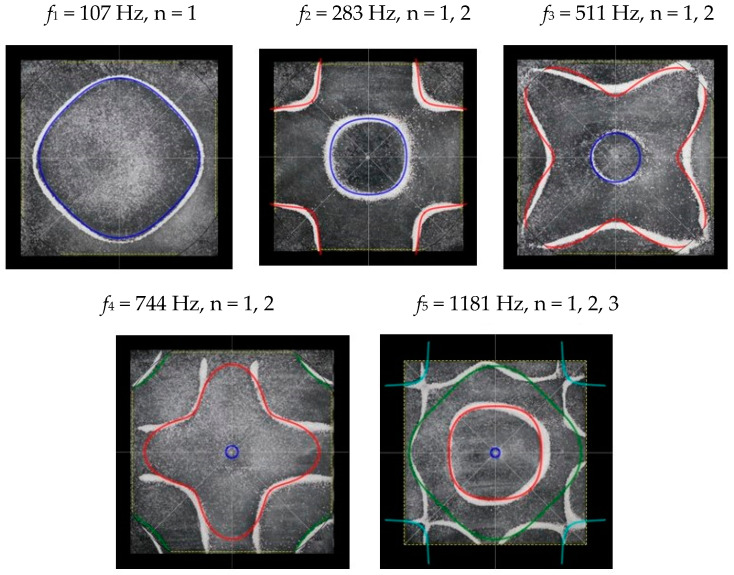
Simulation of nodal line patterns for the larger square plate, a = 24 cm.

**Figure 16 entropy-26-00264-f016:**
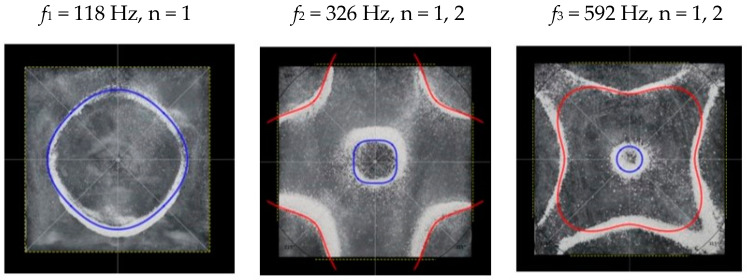
Simulation of nodal line patterns for the smaller square plate, a = 18 cm.

**Figure 17 entropy-26-00264-f017:**
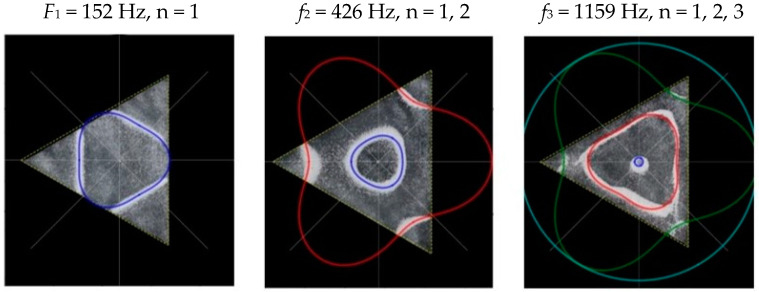
Simulation of nodal line patterns for the larger triangle plate, a = 24 cm.

**Figure 18 entropy-26-00264-f018:**
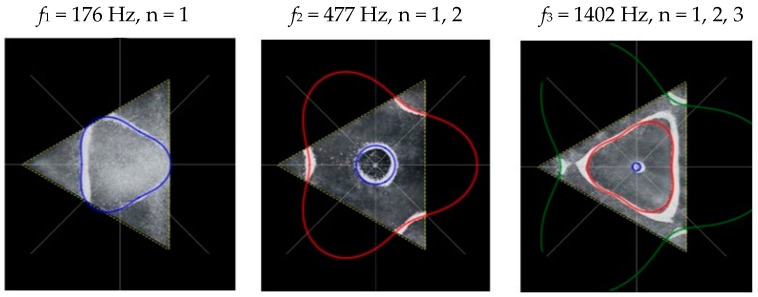
Simulation of nodal line patterns for the smaller triangle plate, a = 18 cm.

**Figure 19 entropy-26-00264-f019:**
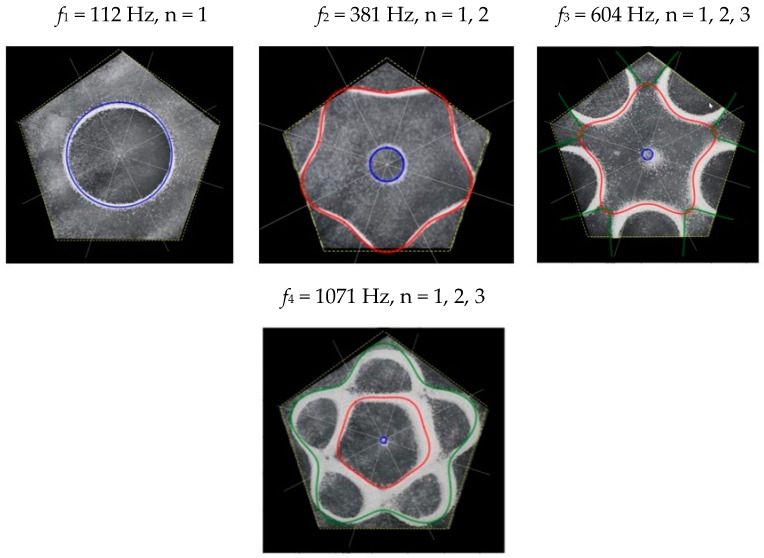
Simulation of nodal line patterns for the larger pentagon plate, a = 14.5 cm.

**Figure 20 entropy-26-00264-f020:**
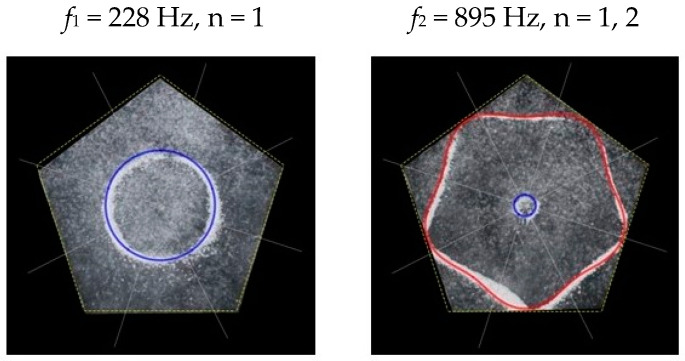
Simulation of nodal line patterns for the smaller pentagon plate, a = 9.5 cm.

**Figure 21 entropy-26-00264-f021:**
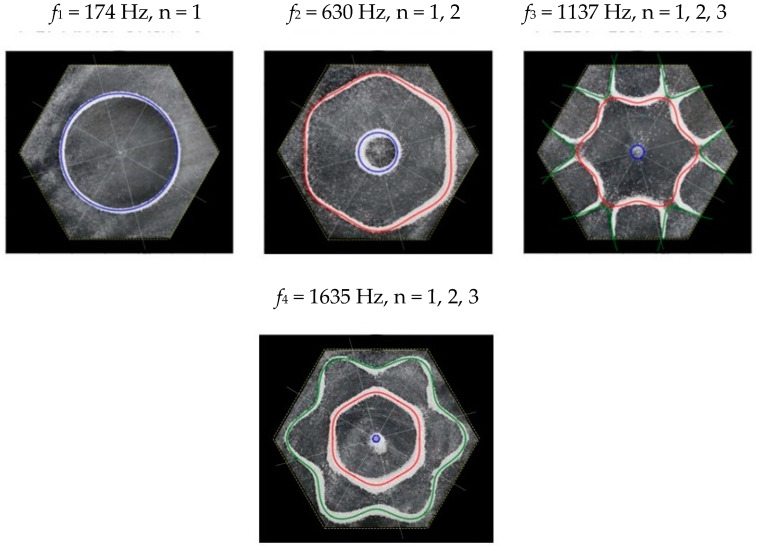
Simulation of nodal line patterns for the larger hexagon plate, a = 12 cm.

**Figure 22 entropy-26-00264-f022:**
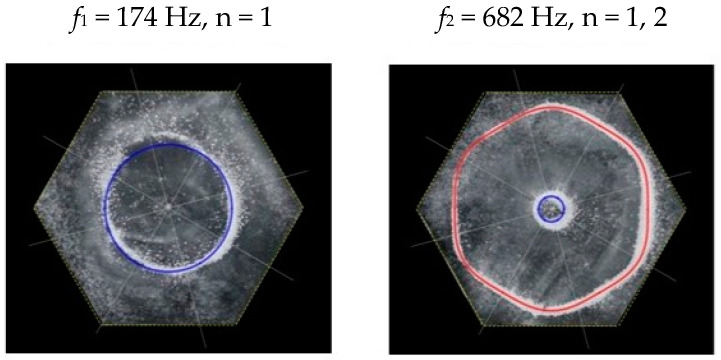
Simulation of nodal line patterns for the smaller hexagon plate, a = 9 cm.

**Table 1 entropy-26-00264-t001:** Resonant frequencies (Hz) found from the impedance peaks for each of the 10 plates.

Shape	Circle		Square		Triangle		Pentagon		Hexagon	
Size (cm)	24	18	24	18	24	18	14.5	9.5	12	9
*f*_1_ (Hz)	158	160	107	118	152	176	112	228	174	174
*f*_2_ (Hz)	582	616	283	326	426	477	381	895	630	682
*f*_3_ (Hz)	1378		511	592	1159	1402	604		1137	
*f*_4_ (Hz)			744				1071		1635	
*f*_5_ (Hz)			1181							

**Table 2 entropy-26-00264-t002:** Coefficients ***A*** and ***C*** of the response function for each of the 10 plates and for each frequency.

			Coefficients							
Shape (cm)	Frequency (Hz)	Velocity (m/s)	A1	A2	A3	A4	C1	C2	C3	C4
circle 24	158	35.6					0.95			
	582	68.2					1.35	1		
	1378	105.0					2.1	1.15	0.75	
circle 18	160	35.8					1.65			
	616	70.2					2.4	2.25		
square 24	107	29.3	−0.1				0.95			
	283	47.6	0.05	−1.85			1.25	0.6		
	511	63.9	0	1.7			1.6	0.7		
	744	77.1	0	−1.2	−1.3		2.7	1.1	0	
	1181	97.2	0	0.2	−0.55	−5.8	2.7	1.45	1.35	−3.65
square 18	118	30.7	−0.05				1.55			
	326	51.1	0	−0.9			2.3	2.2		
	592	68.8	0	0.6			2.5	2.4		
triangle 24	152	34.9	−0.2				1.4			
	426	58.4	0.1	−1.4			1.6	0.8		
	1159	96.3	0	0.7	−2.3		2.7	1.6	−0.8	
triangle 18	176	37.5	−0.2				1.75			
	477	61.8	0	−0.95			2.15	2		
	1402	105.9	0	0.6	−2.8		2.8	2.6	0	
pentagon 14.5	112	29.9	0				1.7			
	381	55.2	0	−0.4			2.3	2.3		
	604	69.5	0	0.7	−2.9		2.8	2.4	2	
	1071	92.6	0	−0.2	1.1		2.9	2.6	2.6	
pentagon 9.5	228	42.7	0				2			
	895	84.6	0	−0.3			2.7	2.35		
hexagon 12	174	37.3	0				1.1			
	630	71.0	0	−2			1.85	1.25		
	1137	95.4	0	0.5	−2.5		2.55	1.4	1.3	
	1635	114.4	0	−0.15	1		2.8	1.6	1.1	
hexagon 9	174	37.3	0				1.9			
	682	73.9	0	−0.1			2.65	2.45		

## Data Availability

Data are contained within the article. The data presented in this study are available upon request from the corresponding author.

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
