# Peer review of "Exploration of Resonant Modes for Circular and Polygonal Chladni Plates"

_entropy, 2024, doi:10.3390/e26030264_

Round 1

Reviewer 1 Report

Comments and Suggestions for Authors

My comments are as follows:

1. The analysis of the manuscript only covered a relatively low frequency range. The authors should clearly specify the numbers related the mode orders. 

2. The work in this manuscript only discussed the simple nodal lines for the circular and polygonal plates. The authors should change the title to clearly describe study topics. 

3.  The authors should discuss how this study is related to "Entropy".   

Author Response

REVIEWER 1

  1. The analysis of the manuscript only covered a relatively low frequency range. The authors should clearly specify the numbers related to the mode orders. 

  1. The work in this manuscript only discussed the simple nodal lines for the circular and polygonal plates. The authors should change the title to clearly describe study topics. 

  1. The authors should discuss how this study is related to "Entropy".   

We thank the referee for the careful evaluation of our manuscript and the insightful comments. We have taken into consideration each point as follows:

  1. The resonant modes for each image are shown in the images (Figures 3-12 and 13-22). However, they have now been included in the table and in the captions. As well a note has been added to distinguish between: the modes referred to as ‘m’; with number of nodal lines ‘n’; and the ‘N’ in the response function, equation which refers to the number of sides of the polygons.

  1. We agree the title could be more informative and modified it as follows:

Exploration of resonant modes for circular and polygonal Chladni plates.

  1. The study is related to entropy in the sense that we are investigating maximal states of entropy of a system i.e. points of equilibrium. We refer to the paper by Shu (2022) who specifically determines the resonant modes in terms of the standard Shannon equation for entropy. This point has been emphasized and included in the introduction and discussion.

Reviewer 2 Report

Comments and Suggestions for Authors

Reference to Chladni plates has been included in the introduction with sufficient number of articles and previous work.  The reference also to the problem of boundary conditions in regular well-known plate vibration problems - such as Kirchoff plates is also included. However, the submission needs to seriously address the description of the problem (objectives of the work) and by providing a comprehensive explanation of the background of Chladni plates, inhomogeneous Helmholtz equations etc. 

Perhaps for a title Scalar Wave Propagation of a Chladni Plate is more appropriate.

Comments on the Quality of English Language

The content of the work is interesting but lacks a coherent thread. Many short paragraphs, disjointed sections and results make the manuscript more difficult to read and flow through the written piece. Please refer to a technical proof reader.

The work is extremely relevant and interesting but challenging to follow.

Author Response

REVIEWER 2

Reference to Chladni plates has been included in the introduction with sufficient number of articles and previous work.  The reference also to the problem of boundary conditions in regular well-known plate vibration problems - such as Kirchoff plates is also included. However, the submission needs to seriously address the description of the problem (objectives of the work) and by providing a comprehensive explanation of the background of Chladni plates, inhomogeneous Helmholtz equations etc. 

Perhaps for a title Scalar Wave Propagation of a Chladni Plate is more appropriate.

The content of the work is interesting but lacks a coherent thread. Many short paragraphs, disjointed sections and results make the manuscript more difficult to read and flow through the written piece. Please refer to a technical proofreader.

The work is extremely relevant and interesting but challenging to follow.

We thank the referee for the careful evaluation of our manuscript and the insightful comments. We have taken into consideration each point as follows:

  1. The objectives of this work have now been more clearly described in the introduction and the discussion.

  1. We agree the title could be more descriptive but would not use the term scaler wave, instead we modified it as follows:

Exploration of resonant modes for circular and polygonal Chladni plates

The writing style has now been addressed and we hope the reviewer will find it has a coherent thread and is easier to follow.

Reviewer 3 Report

Comments and Suggestions for Authors

1. Vibration of plates is described in detail in Vibration of Plates edited by A.W. Leissa. Please include it in your references.

2, The authors explain the Chladni figure using a very simple theoretical formula. However, the explanation of the relationship between these equations and the natural vibration mode of the plate is insufficient.

3. The authors claim that the results of this study have huge potential for a wide range of applications, but they do not explain what those applications are.

4. The research topic is vibration problems in the field of classical mechanics. A detailed explanation of the new discovery is needed.

Author Response

REVIEWER 3

  1. Vibration of plates is described in detail in Vibration of Plates edited by A.W. Leissa. Please include it in your references.

  1. The authors explain the Chladni figure using a very simple theoretical formula. However, the explanation of the relationship between these equations and the natural vibration mode of the plate is insufficient.
  2. The authors claim that the results of this study have huge potential for a wide range of applications, but they do not explain what those applications are.
  3. The research topic is vibration problems in the field of classical mechanics. A detailed explanation of the new discovery is needed.

We thank the referee for the careful evaluation of our manuscript and the insightful comments.  We have taken into consideration each point as follows:

  1. The reference Vibration of Plates edited by A.W. Leissa has now been included in the introduction.

  1. The relationship between the equations presented and the natural vibration mode of the plate has been described in more detail.

  1. The results presented show that for a specific frequency range we are able to successfully determine resonant modes from a simplistic model based on equations of motion. This has huge implications for our understanding on the topic of vibration, which in turn effects how we make calculations and analyse such systems. However, as we move into higher frequencies, the effects of mode mixing become apparent. We hope to extend the model to include these effects.

  1. We appreciate all of the reviewer’s comments, and hope that the amended version gives a more detailed explanation.

Round 2

Reviewer 1 Report

Comments and Suggestions for Authors

Accept in present form

Reviewer 2 Report

Comments and Suggestions for Authors

New version is very acceptable

Comments on the Quality of English Language

English could improve a little, but acceptable in present form.

Reviewer 3 Report

Comments and Suggestions for Authors

The authors revised paper according to reviewer's comments.